# p62-Nrf2-p62 Mitophagy Regulatory Loop as a Target for Preventive Therapy of Neurodegenerative Diseases

**DOI:** 10.3390/brainsci10110847

**Published:** 2020-11-12

**Authors:** Artem P. Gureev, Irina S. Sadovnikova, Natalia N. Starkova, Anatoly A. Starkov, Vasily N. Popov

**Affiliations:** 1Department of Genetics, Cytology and Bioengineering, Voronezh State University, 394018 Voronezh, Russia; ira-ivankina@yandex.ru (I.S.S.); pvn@bio.vsu.ru (V.N.P.); 2Science Department, SUNY Maritime College, New York, NY 10465, USA; natalia.n.starkova@gmail.com; 3Neuroscience Department, Feil Family Brain and Mind Research Institute, Weill Cornell Medicine, New York, NY 10065, USA; ans2024@med.cornell.edu; 4Voronezh State University of Engineering Technologies, 394018 Voronezh, Russia

**Keywords:** mitochondria, Nrf2, p62, mitophagy, regulatory loop, neurodegenerative disease

## Abstract

Turnover of the mitochondrial pool due to coordinated processes of mitochondrial biogenesis and mitophagy is an important process in maintaining mitochondrial stability. An important role in this process is played by the Nrf2/ARE signaling pathway, which is involved in the regulation of the expression of genes responsible for oxidative stress protection, regulation of mitochondrial biogenesis, and mitophagy. The p62 protein is a multifunctional cytoplasmic protein that functions as a selective mitophagy receptor for the degradation of ubiquitinated substrates. There is evidence that p62 can positively regulate Nrf2 by binding to its negative regulator, Keap1. However, there is also strong evidence that Nrf2 up-regulates p62 expression. Thereby, a regulatory loop is formed between two important signaling pathways, which may be an important target for drugs aimed at treating neurodegeneration. Constitutive activation of p62 in parallel with Nrf2 would most likely result in the activation of mTORC1-mediated signaling pathways that are associated with the development of malignant neoplasms. The purpose of this review is to describe the p62-Nrf2-p62 regulatory loop and to evaluate its role in the regulation of mitophagy under various physiological conditions.

## 1. Introduction

Mitochondrial dysfunctions play a key role in a wide range of neurodegenerative diseases, especially Alzheimer’s and Parkinson’s diseases as the most common forms of neurodegenerative diseases [1]. Worldwide, over 46 million people over 65 years old are living with some kind of dementia. Many of them would develop Alzheimer’s disease, eventually [2], a plague of developed countries with a huge negative social-economic impact. Parkinson’s disease is the second most common neurodegenerative disease that affects approximately ten million people worldwide [3]. One of the common biochemical features of these neurodegenerative diseases is neuronal mitochondrial dysfunctions. The latter are very diverse, including overproduction of reactive oxygen species (ROS) [4], oxidative mtDNA damage [5], impaired oxidative phosphorylation as a result of dysfunction of the respiratory complexes [6], ATPase damage [7], a loss of inner membrane integrity [8], a dysfunction of metabolites transport systems [9], and dysfunction of Ca^2+^ metabolism [10]. Each of these dysfunctions can trigger a chain of events that can cause neuronal cell death.

There are few ways of overcoming mitochondrial dysfunctions pharmacologically. The most effective methods are antioxidant protection [11] and manipulating/modulating the turnover of the mitochondrial pool in the cell [12]. The latter includes mitochondrial biogenesis (the formation of new mitochondria), mitophagy (elimination of damaged mitochondria), and fission/fusion processes (mitochondrial dynamics) [13]. Maintaining a dynamic balance between these processes is important for the normal functioning of the structural components of the brain and for the preventive therapy of neurodegenerative diseases [14].

In recent years, the Nrf2 (Nuclear factor erythroid 2-related factor 2) protein has emerged as one of the most promising targets for the therapy of neurodegenerative diseases. It is a transcription factor that regulates the expression of a large number of antioxidant and detoxifying enzymes [15]. However, in the last decade, data have begun to emerge that show its effects on mitochondrial biogenesis and mitophagy [16]. It became clear that Nrf2′s role in maintaining mitochondrial homeostasis is not limited to just antioxidant protection but also extends to the turnover of the mitochondrial pool.

To note, Nrf2 expression and function are also regulated by the same factors that regulate mitochondrial biogenesis. We have previously considered the possibility of the existence of a regulatory loop between Nrf2 and PGC1α in the context of regulation of mitochondrial biogenesis during aging and neurodegeneration [17]. This review is focused on the regulatory loop between Nrf2 and p62, which can potentially be an important target for drugs to cure neurodegenerative diseases.

## 2. Nrf2/ARE Signal Pathway

The transcription factor NF-E2 p45-related factor 2 (Nrf2; gene name *NFE2L2*) regulates the expression of a wide variety of genes that encode proteins with cytoprotective properties, such as antioxidant enzymes, xenobiotic detoxification protein, and anti-inflammatory enzymes, as well as metabolic enzymes and regulators involved in maintaining redox homeostasis [18]. Nrf2 is translocated into the nucleus and binds to the antioxidant response element (ARE) of the promotor in the absence of negative regulators [18]. The regulatory cis-activating element of ARE in the promoter regions of genes is the nucleotide sequence 5′-A (G) TGAC (T) nnnGCA (G) -3′ [19]. There are several variants of the Nrf2 interaction with ARE sequences in the cell nucleus. The most canonical way of activation is the interaction of Nrf2 with basic-leucine zipper (bZip) transcription factors (most often small musculoaponeurotic fibrosarcoma (MAFs)) and CREB-binding protein (CBP) coactivator, which has histone acetyltransferase activity, which leads to changes in the chromatin structure. This allows increasing expression of target genes [20].

Nrf2 is a short-lived protein (about 15 min). In the absence of activating factors, it undergoes ubiquitination and proteasomal degradation. Three ubiquitin ligase systems are known that provide for the degradation of Nrf2. The first to be described was Kelch-like ECH-associated protein 1 (Keap1). Keap1 functions as an adapter protein that mediates the interaction of Nrf2 with the E3 ubiquitin ligase complex Cullin 3 (Cul3), and with RING-box protein 1 (Rbx1), which is required for the interaction of Nrf2 with the ubiquitin ligase system [21]. The second negative regulator of Nrf2 is Glycogen synthase kinase 3 beta (GSK3β), which is able to phosphorylate protein at serine and threonine amino acids. That makes Nrf2 to be recognized by SCF/β-TrCP (SCF is an abbreviation formed from the first letters of the subunits of the complex: Skp1, Cul1, F-box; β-TrCP—β -transducin repeat containing protein). The complex formed by SCF/β-TrCP binds to Cullin 1 (Cul1), which leads to the formation of an ubiquitin ligase complex and subsequent Keap1-independent degradation of Nrf2 [22]. Recently, a third way of negative regulation of Nrf2 was described by the E3 ubiquitin ligase HRD1 [23].

Nrf2 contains seven NRF2-ECH (Neh) domains (Figure 1). The Neh1 domain is required for the formation of a heterodimer with small MAFs and mediates interaction with the ARE sequence of targeted genes [24]. Keap1 binds to the N-terminal Neh2 domain. Thus, Neh2 can be considered the domain responsible for the cytoplasmic localization of Nrf2 [25]. The C-terminal Neh3 domain, as well as the tandem Neh4 and Neh5, provide the transactivating effect of Nrf2 by binding to histone acetyltransferases [26]. In addition, Neh4 and Neh5 mediate interaction with HRD1 [23]. Nrf2 is phosphorylated by GSK3β at the Neh6 domain [27]. The Neh7 domain is responsible for binding to the retinoid X receptor α (RXRα), which can also act as a negative regulator of Nrf2 [28].

Nrf2 inductors are well studied and well described in published studies. These are natural, synthetic, and endogenous quinones, diphenols, phenylenediamines, isothiocyanates, heavy metal ions (Cd, Co, Cu, Au, Hg, Pb), carotenoids, ROS, and chemical compounds that promote their production [18,21,29].

## 3. p62 (Sequestosome 1)

p62 (Sequestosome 1; gene name *SQSTM1*) is a multifunctional cytoplasmic protein that is an important regulatory molecule that functions as a selective autophagy receptor for the degradation of ubiquitinated substrates [30]. Autophagy is an important process that can be divided into two types. Non-selective autophagy occurs in a cell suffering nutritional deficiencies. In this situation, selective autophagy serves to selectively remove organelles in order to regulate their number. Mitophagy is a particular manifestation of selective mitochondrial autophagy [31].

Mitophagy depends on PTEN-induced kinase 1 (PINK1). PINK1 contains a mitochondrial targeting sequence (MTS). In the absence of mitochondrial damage, penetrates into mitochondria through the outer membrane (through the TOM complex) and partially through the inner mitochondrial membrane (through the TIM complex). On the inner membrane, PINK1 undergoes partial cleavage to presenilins-associated rhomboid-like protein (PARL). This form of PINK1 is cleaved by proteases in the mitochondrial matrix [32].

In the damaged mitochondria, the inner membrane is depolarized, which affects TIM-mediated protein import. As a result, PINK1 protein does not enter the mitochondrial matrix, where it is usually degraded. Therefore, the PINK1 protein accumulates on the outer mitochondrial membrane. It leads to the activation of PARKIN, which is the cytosolic E3-ubiquitin ligase. PARKIN can ubiquitinate a number of proteins on the outer mitochondrial membrane, which triggers mitophagy. PARKIN promotes K63-linked polyubiquitination of the mitochondrial substrate and recruits ubiquitin- and LC3-binding protein p62 into mitochondria [33]. The p62 acts as an adapter molecule that directly interacts with ubiquitinated molecules on the autophagosome. Ablating p62 completely blocks the clearance of damaged mitochondria [34]. Thus, activation of the PINK1/PARKIN/p62 axis plays an important role in the selective elimination of damaged mitochondria, which is essential for maintaining their quality control. It should be noted, that some recent data suggest that p62-mediated ubiquitination and mitophagy can also be carried out in the PINK1/PARKIN-independent pathway [35].

The p62 protein is made of multiple domains that provide a wide range of functions (Figure 1). The PB1 domain (Phox and Bem1p) is responsible for interacting with the autophagy receptor NBR1 and with a number of protein kinases (ERK, MEKK3, MEK5, and aPKCs). This domain is also responsible for protein di- and multimerization. The ZZ-type zinc finger domain is responsible for binding to RIP1 (receptor-interacting serine threonine kinase 1). The TB domain (TNF receptor-associated factor 6 (TRAF6) binding domain) contains the E3 binding site of the ubiquitin-protein ligase TRAF6. The LIR domain (C-terminal LC3- interacting region) and UBA domain (ubiquitin-associated domain) link the autophagic machinery to ubiquitinated protein substrates. Finally, KIR (Keap-interacting region) binds Keap1 and induce Nrf2 nuclear translocation [36].

## 4. p62-Nrf2 Regulatory Loop

The ability of p62 to activate Nrf2 was first described in 2010 by at least three research groups. The first was done by Komatsu et al. (2010), who demonstrated that p62 interacts with the Nrf2-binding site in Keap1, and that p62 accumulation results in an activation of Nrf2 [37]. At about the same time, Lau et al. (2010) presented the data that there is a direct interaction between p62 and Keap1. Accumulation of p62 sequesters Keap1 into aggregates, resulting in an inhibition of Keap1-dependent ubiquitination and subsequent degradation of Nrf2 [38]. At the same time, by means of immunopurification and mass spectrometry, an interaction between Keap1 and p62 was shown [39]. In the same year, the Komatsu group demonstrated that a simultaneous knockout of p62 and Nrf2 resulted in autophagy suppression, which also indirectly indicates the relationship between p62 and Nrf2 and the fact that the accumulation of p62-Keap1 aggregates leads to constitutive activation of Nrf2 [40]. However, it was shown that these aggregates are observed in more than 25% of human hepatocellular carcinomas. It is noteworthy that it is an increase in p62 expression, rather than Keap1 mutants, that causes malignant growth [41].

Jain et al. (2010) discovered ARE sequences in the promoter region of the gene encoding p62 and verified that Nrf2 binds to this *cis*-element in vivo and in vitro [42] (Figure 1). It has been shown that PMI (P62-mediated mitophagy inducer) and sulforaphane (Nrf2 inducer) are able to activate p62 expression via the Nrf2/ARE signaling pathway [43]. Recently, Liao et al. (2019) proved the existence of a positive feedback loop that is activated by cisplatin, which induces oxidative stress in an acute kidney injury model. The p62 knockdown significantly decreased Nrf2 protein expression, which was accompanied by an increase in oxidative stress. In turn, Nrf2 knockdown significantly reduced the cisplatin-induced expression of p62, and this caused a disruption in autophagosome formation [44].

However, there are some data suggesting that p62 may not activate Nrf2, but suppress its activity. There is a splicing variant of p62 that is lacking the last half of the KIR domain, which interacts with Keap1. As a result, there is ubiquitination of Nrf2 that is leading to its degradation by the 26S proteasome. This suppresses the expression of Nrf2 target genes [45]. However, the role of this p62 splicing variant in the regulation of mitophagy and its consequences for the pathogenesis of neurodegenerative diseases requires further studies.

## 5. Physiological Role of p62-Nrf2 Regulatory Loop

Both separately and together, p62 and Nrf2 signaling pathways are firmly associated with cell survival [46]. On the one hand, p62 and Nrf2 signaling pathways can protect a tissue from degeneration, which is especially important considering neurodegeneration. On the other hand, activation of these signaling pathways can lead to the development of oncological processes. A review by Katsuragi et al. (2016) discussed in detail the role of p62 and Nrf2 signaling pathways in the pathogenesis of hepatocellular carcinoma [47]. p62 and Nrf2 include activation phosphatidylinositol 3-kinase (PI3K)-Akt pathway and mammalian target of rapamycin complex 1 (mTORC1). Nrf2 positively regulates MTOR expression [48]. p62 interacts with molecules, which interact with mTOR and forms mTORC1. The p62 activation is a crucial step for mTOR activation [49]. The mTORC1 plays a dual role in the regulation of cellular processes. It is definitely necessary for long-term potentiation via regulation of protein synthesis [50]. On the other hand, mTOR promotes cell growth signaling, and its mutations were identified in several types of human cancer [51].

However, the p62-Keap1-Nrf2 axis promotes malignancy of hepatocellular carcinoma through enhancing UDP-glucuronate and glutathione production, which can promote hepatocellular carcinoma growth [52]. There are several reasons why p62 and Nrf2, which are thus important for mitophagy, may cause hepatocellular carcinoma. The reasons are mutations in the *NFE2L2* and *KEAP1* genes [53], chronic inflammation [54], which cause constitutive Nrf2 activation, and stable overexpression of p62.

In either case, the p62-Nrf2 regulatory loop is attractive for pharmacological intervention because it appears to be a good target for developing compounds that are aimed to suppress neurodegenerative processes. E.g., an accumulation of misfolded peptides (α-synuclein in Parkinson’s disease, a-beta and tau fibrils in Alzheimer’s disease, et cetera). In so far as ubiquitin plays a critical role in the elimination of misfolded and aggregated protein molecules, p62 is a good target for modulating proteasomal pathways [55]. A decrease in expression or inactivation of p62 gene in mice had caused some of the symptoms associated with Alzheimer’s disease (loss of working memory) and resulted in hyperphosphorylated tau, neurofibrillary tangles, and neurodegeneration [56]. Similar results were observed for Alzheimer’s disease rat model by injecting β-amyloid protein into the hippocampus, where p62 expression was reduced [57]. An increase in p62 expression resulted in a decrease in Aβ level and improved cognitive ability in APP/PS1 mice (a mouse model of Alzheimer’s disease) [58]. These results indicate that an increase in p62 expression may be a target to reduce the Aβ level and cognitive impairment.

At present, there are no compounds that modulate p62 expression; these have to be developed. However, a large number of Nrf2 activators have been described (Table 1). At the moment, only dimethyl fumarate is approved by the Food and Drug Administration (FDA) for the treatment of neurodegenerative disease (multiple sclerosis) [59]. In addition to dimethyl fumarate, Nrf2 activators such as curcumin, resveratrol, sulforaphane, masatinib, methylene blue, omaveloxolone, tideglusib, Dl-3-n-butylphthalide ide, ALKS-8700, benfotiamine, and ketogenic diet undergoing clinical trials for treating various neurodegenerative disease such as Alzheimer’s disease, Parkinson’s disease, Huntington’s disease, Friedreich’s ataxia, multiple sclerosis, amyotrophic lateral sclerosis, cataract, schizophrenia, bipolar disorder, mild cognitive impairment, depression, autism, obstructive sleep apnea, etc. [59]. A number of compounds of Nrf2 activators have been studied in animal and cellular models of Alzheimer’s and Parkinson’s disease. Among the most promising compounds (besides compounds undergoing clinical trials) for the treatment of Parkinson’s disease are carnosic acid, monomethyl fumarate, salidroside, β-ecdysterone, pinostrobin, berberine, vildagliptin, glaucocalyxin B, fasudil, protocatechuic acid, chrysin, hypoestoxide, α-Asarone [16,60]. Among the most promising compounds (besides compounds undergoing clinical trials) for the treatment of Alzheimer’s disease are carnosic acid, gypenoside XVII, eriodictyol, hesperidin, puerarin, orientin, antroquinonol, sodium hydrosulfide, vanillic acid, methysticin, 3H-1,2-dithiole-3-thione, mini-GAGR, allicin, triterpenoid CDDO-methylamide (CDDO-MA) [60,61]. Other triterpenoids such as CDDO-ethyl amide (CDDO-EA) and CDDO-trifluoroethyl amide (CDDO-TFEA) improve the behavioral phenotype in a model of Huntington’s disease [62] (Table 1). Unfortunately, none of these compounds are in clinical trials as of now. The major reason for that is—in our understanding—is that the signaling mechanisms underlying the interaction of mitochondria with neuronal metabolism are insufficiently studied, as well as the mechanisms that actually control the survival of neurons in their native location (human brain).

## 6. Conclusions

The value of Nrf2 activators is limited not only to their ability to reduce oxidative stress, which has been repeatedly discussed [95,96,97]. Oxidative stress is not only a cause of mitochondrial dysfunction but a consequence of a disruption of mitochondrial quality control [98]. An imbalance occurs when there is a violation of the coordination of mitochondrial biogenesis and mitophagy [99]. When biogenesis is suppressed, and mitophagy is activated, an energy deficit occurs. When mitophagy is suppressed, and mitochondrial biogenesis is activated, a large number of damaged mitochondria can accumulate in the cell, which will produce a lot of ROS, but at the same time not fully meet the energy requirements due to the non-functional respiratory chain [99]. Nrf2 is capable of forming regulatory loops that are involved in the regulation of mitochondrial biogenesis. There is evidence that Nrf2 increases the expression of peroxisome proliferator-activated receptor-gamma coactivator 1-alpha (PGC-1α) (master regulator mitochondrial biogenesis) and nuclear respiratory factor (NRF1), which are directly involved in the regulation of mtDNA transcription. PGC-1α, in turn, deactivates GSK3β via p38 [17]. Another loop involves interaction with p62, which forms a loop with Nrf2 by inactivating Keap1. However, the regulation of mitophagy by Nrf2 is not limited to this. There is evidence that Nrf2 regulates the expression of Pink1, which plays a key role in mitophagy induction (Figure 2). It is important that Nrf2 activation does not lead to an imbalance in the direction of mitophagy or mitochondrial biogenesis but rather to maintain a dynamic balance, which is essential for mitochondrial stability.

## Figures and Tables

**Figure 1 brainsci-10-00847-f001:**
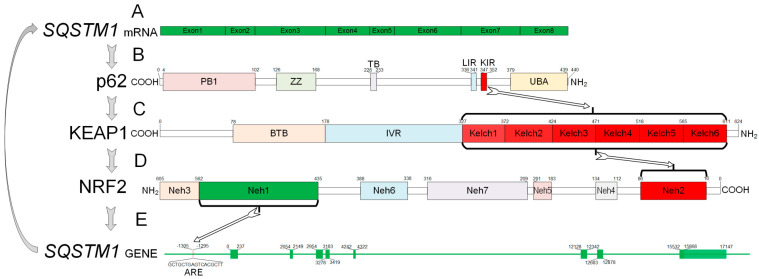
Scheme of p62-Nrf2 loop with protein and gene structure. KIR of p62 interacts with the Kelch repeat of KEAP1 protein and inhibits its activity. Kelch repeat interacts with the Neh2 domain of the Nrf2 protein and inactivates it. Free Nrf2 can interact with the ARE region of the *SQSTM1* gene and increase its expression. A. mRNA structure of *SQSTM1* of humans. mRNA contains 1352 protein-coding bp and includes 8 exons. B. Domain structure of p62 protein of human (440 aa). Phox and Bem1p (PB1)—4–102 aa; ZZ-type zinc finger domain (ZZ)—126–168 aa; TRAF6 binding domain (TB)—228–233 aa; LC3 interacting region (LIR)—336–341 aa; KEAP1 interacting region (KIR)—347–352 aa; ubiquitin-associated domain (UBA)—379–439 aa. C. Domain structure of KEAP1 protein of human (624 aa). BTB domain (BTB)—78–178 aa; intervening region (IVR)—179–327 aa; Kelch repeat (Kelch1–Kelch6)—327–611 aa. D. Domain structure of KEAP1 protein of human (605 aa). Neh domains. Neh2—16–86 aa; Neh4—112–134 aa; Neh5—183–201 aa; Neh7—209–316 aa; Neh6—338–388 aa; Neh1—435–562 aa; Neh3—563–605 aa. E. *SQSTM1* gene structure of humans. Gene contains 8 exons. 1st exon (0–237 bp); 2nd exon (2054–2149 bp); 3rd exon (2954–3183 bp); 4th exon (3278–3419 bp); 5th exon (4249–4322 bp); 6th exon (12128–12342 bp); 7th exon (12683–12878 bp); 8th exon (15532–17147 bp). *SQSTM1* gene contains ARE region at (−1306)–(−1295) bp.

**Figure 2 brainsci-10-00847-f002:**
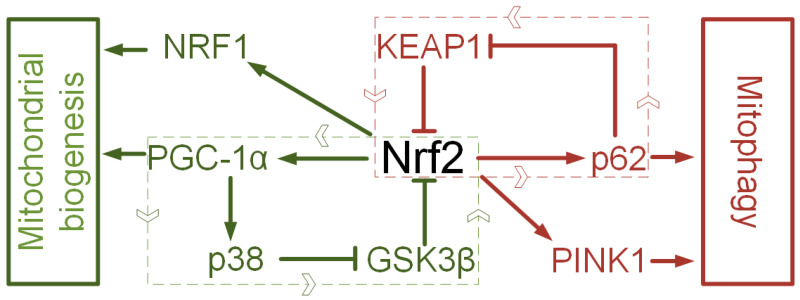
Two regulatory loops for mitochondria turnover. Nrf2—PGC1α—p38—GSK3β—Nrf2 loop and Nrf2—NRF1 interaction regulate of mitochondrial biogenesis. Nrf2—p62—KEAP1—Nrf2 loop and Nrf2—PINK1 interaction regulate of mitochondrial biogenesis.

**Table 1 brainsci-10-00847-t001:** Nrf2 activators undergoing clinical trials and research in the cells and animal models of neurodegenerative diseases.

Compound	Disease	Research Type	Reference
Curcumin	Schizophrenia	Clinical trial. Phase I	NCT02104752 *
	Alzheimer’s disease	Clinical trial. Phase I/II	NCT00164749
	Schizophrenia	Clinical trial. Phase IV	NCT02298985
	Mild cognitive impairment	Clinical trial. Phase II	NCT01811381
	Major depression	Clinical trial. Phase IV	NCT01750359
	Parkinson’s disease	Rat model. Rotenone-injured	[63]
Resveratrol	Alzheimer’s disease	Clinical trial. Phase II	NCT01504854
		Clinical trial. Phase III	NCT00743743
	Huntington’s disease	Clinical trial. Phase III	NCT02336633
	Friedreich’s ataxia	Clinical trial. Phase II	NCT03933163
Sulforaphane	Schizophrenia	Clinical trial. Phase II	NCT02880462
		Clinical trial. Phase II	NCT02810964
		Clinical trial. Phase II	NCT01716858
	Autism	Clinical trial. Phase II	NCT01474993
		Clinical trial. Phase II	NCT02909959
		Clinical trial. Phase II	NCT02677051
		Clinical trial. Phase III	NCT02654743
		Clinical trial. Phase I/II	NCT02561481
	Alzheimer’s disease	Clinical trial. Recruiting	NCT04213391
	Parkinson’s disease	Cell models. Mice models. 6-OHDA treated.	[64]
Dimethyl fumarate	Multiple sclerosis	Clinical trial. FDA Approved in 2013	NCT02047097, NDA 204063 **
	Obstructive sleep apnea	Clinical trial. Phase II	NCT02438137
	Alzheimer’s disease	Mice models. P301L mice	[65]
	Parkinson’s disease	Mice models. MPTP treated	[66]
	Huntington’s disease	Mice models. R6/2 and YAC128 mice	[67]
Masatinib	Multiple sclerosis	Clinical trial. Phase III	NCT01433497
	Alzheimer’s disease	Clinical trial. Phase II/III	NCT01872598
	Amyotrophic lateral sclerosis	Clinical trial. Phase II	NCT02588677
Methylene blue	Alzheimer’s disease	Clinical trial. Phase II	NCT02380573
	Bipolar disorder	Clinical trial. Phase III	NCT00214877
	Parkinson’s disease	Mice models. MPTP treated	[68]
Omaveloxolone	Friedreich’s ataxia	Clinical trial. Phase II	NCT02255435
	Cataract	Clinical trial. Phase II	NCT02128113
Tideglusib	Autism	Clinical trial. Phase II	NCT02586935
	Alzheimer’s disease	Clinical trial. Phase II	NCT01350362
Ketogenic diet	Alzheimer’s disease	Clinical trial. Phase II	NCT04466735
		Clinical trial. Not Applicable	NCT03690193
Dl-3-n-butylphthalide ide	Alzheimer’s disease	Clinical trial. Not Applicable	NCT02711683
ALKS-8700	Multiple sclerosis	Clinical trial. Phase III	NCT02634307
Benfotiamine	Alzheimer’s disease	Clinical trial. Phase II	NCT02292238
Carnosic acid	Alzheimer’s disease	Mice models.(hAPP)-J20 and 3×Tg-AD mice	[69]
	Parkinson’s disease	Cell models. Paraquat treated SH-SY5Y cells	[70]
ITH12674	Brain ischemia	Culture of rat cortical neurons	[71]
Monomethyl fumarate	Parkinson’s disease	Mice models. MPTP treated	[66]
Salidroside	Parkinson’s disease	Cell models. MPP^+^/MPTP treated	[72]
β-Ecdysterone	Parkinson’s disease	Cell models. MPP^+^ treated	[73]
Pinostrobin	Parkinson’s disease	Cell models. MPTP treated SH-SY5Y cells	[74]
Berberine	Parkinson’s disease	Zebrafish models. 6-OHDA treated	[75]
Vildagliptin	Parkinson’s disease	Rat models. Rotenone treated	[76]
Glaucocalyxin B	Parkinson’s disease	Rat models. Lipopolysaccharide-injected	[77]
Fasudil	Parkinson’s disease	Mice models. MPTP treated	[78]
Protocatechuic acid	Parkinson’s disease	Cell models. 6-OHDA treated PC12 cells	[79]
Chrysin	Parkinson’s disease	Cell models. 6-OHDA treated PC12 cells	[79]
Hypoestoxide	Parkinson’s disease	Mice models. mThy1-α-syn transgenic mice	[80]
α-Asarone	Parkinson’s disease	Mice models. MPTP treated	[81]
Gypenoside XVII	Alzheimer’s disease	Cell models. Aβ treated	[82]
Eriodictyol	Alzheimer’s disease	Cell models. Aβ treated	[83]
Hesperidin	Alzheimer’s disease	Mice models. APP/PS1 mice	[84]
Puerarin	Alzheimer’s disease	Mice models. APP/PS1 mice	[85]
Orientin	Alzheimer’s disease	Mice models. Aβ injected	[86]
Antroquinonol	Alzheimer’s disease	Mice models. Aβ injected	[87]
Sodium hydrosulfide	Alzheimer’s disease	Mice models. APP/PS1 mice	[88]
Vanillic acid	Alzheimer’s disease	Mice models. Aβ injected	[89]
Methysticin	Alzheimer’s disease	Mice models. APP/PS1 mice	[90]
3H-1,2-dithiole-3-thione	Alzheimer’s disease	Mice models. Tg2576 mice	[91]
Mini-GAGR	Alzheimer’s disease	Mice models. 3xTg-AD mice	[92]
Allicin	Alzheimer’s disease	Rat models. Tunicamycin-injected	[93]
CDDO-MA	Alzheimer’s disease	Mice models. Tg19959 mice	[94]
CDDO-EA	Huntington’s disease	Mice models. N171-82Q mice	[62]
CDDO-TFEA	Huntington’s disease	Mice models. N171-82Q mice	[62]

* NCT—national clinical trial; ** NDA—new drug approval.

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
