# Peer review of "p62-Nrf2-p62 Mitophagy Regulatory Loop as a Target for Preventive Therapy of Neurodegenerative Diseases"

_brainsci, 2020, doi:10.3390/brainsci10110847_

Round 1
Reviewer 1 Report
The manuscript "p62-Nrf2-p62 mitophagy regulatory loop as a target for preventive therapy of neurodegenerative diseases" by Gureev et al., is well written, however, evidence regarding the p62-Nrf2-p62 mitophagy regulatory loop as a "preventive measure" needs to be elaborated in a separate heading for various/selected neurodegenerative disorders.
Author Response
We thank you for your appropriate and important comment. We have expanded the section on Nrf2 activators as potential drugs for neurodegenerative diseases. In this section, we have mentioned not only Alzheimer's and Parkinson's, but also some other diseases, such as Huntington’s disease, Friedreich's ataxia, multiple sclerosis, amyotrophic lateral sclerosis, schizophrenia, bipolar disorder, mild cognitive impairment etc. We have described compounds that are undergoing clinical trials, and those that are being currently studied in animals and cell models. We have added a summary table to this section, where we’ve classified the basic information on the known Nrf2 activators. We hope that with this addition, as suggested by our reviewers, the review has become more comprehensive regarding the role of these transcription factors in the treatment of neurodegenerative diseases.
Sincerely, authors.

Reviewer 2 Report
Gureev et al. reviewed the p62-Nrf2-p62 loop in regulating mitophagy and reviewed the studies of the loop in neurodegenerative diseases. The authors introduced the Nrf2 and p62 protein structures, the working mechanisms of Nrf2 and p62 in regulating mitophagy by interacting with other proteins, and the p62-Nrf2 regulatory loop. At last, the physiological role of p62-Nrf2 regulatory loop including its role in cancer and neurodegenerative diseases has been summerized. This review article clearly introduced the p62-Nrf2 loop and its role in mitophagy and will help researchers to understand this loop in neurodegenerative diseases. There is some suggestion to improve the article before publication:
The authors can list a table about the protective effects of p62-Nrf2 in various neurodegenerative diseases. In the current version, there is only one paragraph introducing it as a potential therapeutic target. More studies on this loop could be included and summarized in a table.
Author Response
Thank you very much for the valuable advice on adding a table to the manuscript. We tried to collect and summarize as much information as possible about Nrf2 activators in a table. This table contains the name of the compound, its field of application (a specific neurodegenerative disease), the stage of research (clinical trials or research in the animals or cell models), reference (NCT or NDA where approved by FDA for clinical research, and a reference to the original article for other research).
Sincerely, authors.
